# Improving Sentence Representations via Component Focusing

**Xiaoya Yin [1,\*], Wu Zhang [1], Wenhao Zhu [1,\*], Shuang Liu [1] and Tengjun Yao [2]**

[1]   School of Computer Engineering and Science, Shanghai University, Shanghai 200444, China; wzhang@shu.edu.cn (W.Z.); ls18365633667@163.com (S.L.)

[2]   The 36th Institute China Electronics Technology Group Corporation, Jiaxing 314000, China; yaotj@jec.com.cn

\*   Correspondence: shu_yxy@163.com (X.Y.); whzhu@shu.edu.cn (W.Z.); Tel.: +86-1881-723-1159 (X.Y.)



**Featured Application: Natural language processing (NLP) is a crossing domain of computer science, artificial intelligence, and linguistics that focuses on the interaction between computers and human (natural) languages. NLP faces many challenges, including sentence representation. Appropriate sentence representation significantly improves the efficiency of NLP tasks such as text comprehension, text classification, machine translation, and information extraction.**

**Abstract:** The efficiency of natural language processing (NLP) tasks, such as text classification and information retrieval, can be significantly improved with proper sentence representations. Neural networks such as convolutional neural network (CNN) and recurrent neural network (RNN) are gradually applied to learn the representations of sentences and are suitable for processing sequences. Recently, bidirectional encoder representations from transformers (BERT) has attracted much attention because it achieves state-of-the-art performance on various NLP tasks. However, these standard models do not adequately address a general linguistic fact, that is, different sentence components serve diverse roles in the meaning of a sentence. In general, the subject, predicate, and object serve the most crucial roles as they represent the primary meaning of a sentence. Additionally, words in a sentence are also related to each other by syntactic relations. To emphasize on these issues, we propose a sentence representation model, a modification of the pre-trained bidirectional encoder representations from transformers (BERT) network via component focusing (CF-BERT). The sentence representation consists of a basic part which refers to the complete sentence, and a component-enhanced part, which focuses on subject, predicate, object, and their relations. For the best performance, a weight factor is introduced to adjust the ratio of both parts. We evaluate CF-BERT on two different tasks: semantic textual similarity and entailment classification. Results show that CF-BERT yields a significant performance gain compared to other sentence representation methods.

**Keywords:** natural language processing; sentence representation; sentence embedding; component focusing; semantic textual similarity

---

## 1. Introduction

Much progress has been made in learning semantically meaningful distributed representations of individual words, such as Word2Vec [1], GloVe [2], and ELMo [3]. On the other hand, much remains to be done to obtain satisfying representations of sentences, also known as sentence embeddings. The main idea of sentence embedding is to encode sentences into fixed-sized vectors. The sentence representations are usually used as features for subsequent machine learning tasks or pre-training in the context of deep learning. The applications of sentence representations are many, including text classification [4], sentence similarity [5], question-answering [6], and information retrieval [7], to name

a few. Appropriate sentence representations can significantly improve the performance of natural language processing (NLP) tasks.

Among all sentence representations methods, the vector average method [8] is the easiest and most popular. Although more complex sequential networks such as LSTMs (Long Short-Term Memory) [9] or convolutional [10] networks may yield better performance, the improvement is not as significant as expected. Sometimes, the trade-off between efficiency and performance tips the balance in favor of simpler models like vector averaging. Transformer [11] is a new network structure to replace the recurrent neural network (RNN) and convolutional neural network (CNN). It directly obtains global information unlike the RNN, which requires gradual recursion to obtain global information, and unlike the CNN, which only obtains local information. Transformer outperforms RNN and CNN in NLP tasks such as machine translation and can also run in parallel, many times faster than RNN. Bidirectional encoder representations from transformers (BERT) [12] has attained much attention nowadays because it utilizes the transformer network to obtain state-of-the-art results in a wide array of NLP tasks. A large disadvantage of BERT network structure is that no independent sentence embeddings are computed, which makes it difficult to derive sentence embeddings from BERT.

The standard sentence methods do not adequately address some linguistic properties, which are important factors for producing appropriate sentence representations. A sentence is composed of different components, such as the subject, predicate, object, attributive, adverbial phrase, and complements. In the process of constructing a sentence, each sentence component does not function exactly the same. Of all the components, the subject, predicate, and object serve the most crucial roles because they represent the primary meaning of a sentence, while the others are seen as noisy components because they are less informative. Thenmozhi et al. [13] analyzed the text similarity between the search term and the subject-predicate-object information of the retrieved text sentence, which can improve the precision of a semantic search. Additionally, words within a sentence are also related to each other, not only just by their positions but also syntactic relations. In a prior work, Levy et al. [14] generalized that changing a skip-gram model from a linear context to a dependency-based syntactic relations context can lead to better performance in word similarity. Ma et al. [15] proposed a dependency-based convolution approach, making use of tree-based n-grams rather than sequential ones, thus utilizing nonlocal interactions between words to improve sentence modeling baselines on four sentiment and question classification tasks.

Therefore, we developed component focusing BERT (CF-BERT), a modification of the pre-trained BERT network that uses a Siamese network structure to derive semantically meaningful sentence embeddings via component focusing. The CF-BERT divides a sentence representation into two parts: a basic sentence part refers to the complete sentence, and the component-enhanced part, which contains the crucial sentence information (primarily from the subject, predicate, and object of a sentence) and the subject-predicate-object syntactic relations. The basic part contains over-sufficient information of a sentence, the same as the traditional method. While the component-enhanced part takes advantage of the relevant information and reduces the impact of noisy words on the sentence meaning.

To be specific, we adopt syntactic dependency parsing to acquire the component-enhanced part of a sentence. According to the dependency parsing of a sentence, shown in Figure 1, we can directly obtain the dependency relations between words and derive components such as subject, predicate, and object secondhand.

The CF-BERT reads sentences with different lengths to generate fixed-length representations of both parts. While generating the sentence embeddings, the basic component occupies the dominant position and the component-enhanced part plays a supplement role. Hence, a weight factor is introduced to adjust the ratio of the embeddings of two parts to generate the complete sentence representation. The grid search method is used to get a weight factor to achieve optimal sentence representation. This sentence representation then implements a pooling strategy, and then gets the final vector for similarity calculation and entailment classification, with significant results to other sentence representation methods.

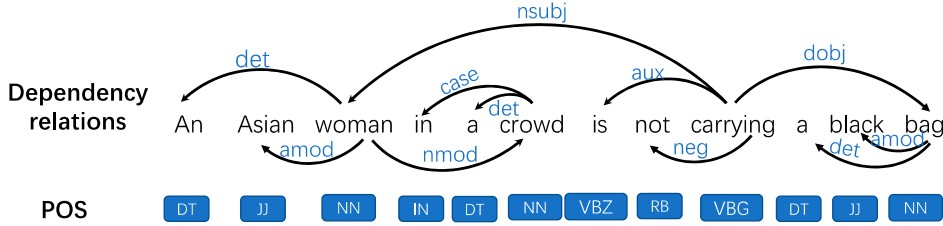

**Figure 1.** An example of dependency parsing. The dependency relation between two words is represented by a directed arc. Meanwhile, the part-of-speech (POS) of each word is also obtained.

The key contribution of this paper is to propose a novel sentence representation model focused on the crucial components of a sentence with their syntactic relations. The addition of a component-enhanced part in the basic representation can enhance the useful information of the sentence representation and alleviate the noisy words, with significant and consistent improvement in the downstream NLP tasks.

In the next section, we present a summary of related studies on sentence representations. Our paper focuses on the similarity task (Sentences Involving Compositional Knowledge Semantic Relatedness (SICK-R) and Semantic Texture Similarity Benchmark (STS-B)), and then the sentence representations obtained by training the similarity task SICK-R are directly used for the text classification task (Sentences Involving Compositional Knowledge Entailment (SICK-E)). The design of the method and training details are described in Section 3 and only focus on textual similarity. Then, we explain the data sources, experimental details, experimental results, evaluation, and discussion of the experiment in Sections 4 and 5. Particularly in Section 4.4, we evaluate the broader utility of our sentence representations on entailment classification.

## 2. Related Work

Learning representations of sentences, also called sentence embeddings, are a well-studied area with dozens of proposed methods.

The most popular and easiest way to generate a sentence embedding is simply by averaging word embeddings [8] in spite of their apparent disregard for syntactic structure. An improved method uses a weighted average of word vectors. Arora et al. constructed a sentence embedding called SIF (smooth inverse frequency) [16] as a sum of pre-trained word embeddings, weighted by reverse document frequency, and subtracted a vector based on principal components from the sentence vectors.

In terms of neural networks, convolutional neural network (CNN) [17–19] was recently applied efficiently for semantic composition. This technique uses convolutional filters to capture local dependencies in terms of context windows and applies a pooling layer to extract global features. Many other works apply a recurrent neural network (RNN) and its powerful variants, such as LSTMs, to learn better sentence representations, and they have achieved substantial success in text categorization [20], machine translation [21], etc. Sent2Vec [22] learns n-gram features in a sentence to predict the center word from the surrounding context. A sequential neural sentence encoder, like Skip-thought [23], trains an encoder-decoder architecture that can predict the surrounding sentences. The sequential neural sentence encoder InferSent [24] uses labeled data from the Stanford Natural Language Inference dataset (SNLI) [25] and Multi-Genre NLI dataset [26] to train a Siamese BiLSTM (Bi-directional Long Short-Term Memory) network with max-pooling over the output to generate the sentence encoder. Conneau et al. showed that InferSent consistently outperforms unsupervised methods like Skip-thought. Universal Sentence Encoder [27] trains a transformer network and augments unsupervised learning. The model is first trained on a large scale of unsupervised data from Wikipedia and forums and then trained on the SNLI dataset.

Recent works highlight the strength of Transformer [11] architecture on sequence tasks. BERT [12] is a pre-trained transformer network, which breaks 11 NLP tasks with state-of-the-art results, including machine reading comprehension, sentence classification, and sentence-pair similarity. One big

drawback of BERT network structure is that independent sentence embeddings cannot be computed directly, which makes it challenging to derive sentence embeddings from BERT. To overcome this restriction, researchers feed single sentences through BERT and then derive a fixed-sized vector by either averaging the outputs (similar with average word embeddings) or by using the output of the first token (the [CLS (classification)] token) [28,29]. The above two methods are also provided by the popular service called bert-as-a-service-repository. However, it produces out-of-the-box rather bad sentence embeddings.

An account of previous works [30] found that the NLI (including SNLI and Multi-Genre NLI) datasets are suitable for training sentence embeddings. Sentence-BERT (SBERT) [31] uses a Siamese network structure to fine-tune the pre-trained BERT network first by NLI and then by specific NLP tasks to derive semantically meaningful sentence embeddings. Using a similarity measure like cosine-similarity or Manhattan/Euclidean distance, semantic textual similarity between two sentence embeddings are calculated.

Additional works have also exploited linguistic structures such as parse and dependence trees to improve sentence representations. Chen et al. [32] used a neural network to develop a transition-based greedy model that remarkably improves the accuracy and speed of dependency parsers. Tai et al. [33] designed a dependency tree-structured LSTM for modeling sentences. This model outperforms the linear chain LSTM in semantic textual similarity tasks. Subsequently, researchers have applied dependency to CNNs and RNNs in the tasks, including relation classification [34], Chinese word segmentation [35], translation [36], etc. DisSent [37] shows that dependency parsing and rule-based rubrics can curate a high-quality sentence relation task by leveraging explicit discourse relations. All these models can potentially encode richer semantic and syntactic features from sentence structures with dependency parsing.

## 3. Methods

### 3.1. Model Architecture

Our model CF-BERT aims to focus on crucial components and syntactic relations of a sentence to get a more powerful sentence representation that yields better performance in downstream NLP tasks. As mentioned before, the sentence representation consists of two parts, and to make it easy to describe a sentence representation that consists of two parts, we give the raw sentence text of the two parts names: $S_{basic}$ and $S_{cf}$. The basic sentence part $S_{basic}$, contains the complete sentence information, and the component-enhanced sentence part $S_{cf}$, contains the crucial (primarily from the subject, predicate, and object of a sentence) sentence information. $S_{basic}$ essentially equals to the raw text of a sentence in NLP datasets which cover the global information of a sentence, while $S_{cf}$ enhances semantic information by keeping only crucial information and removing meaningless or noisy words. The example sentence results of basic component $S_{basic}$ and component-enhanced $S_{cf}$ are listed in Table 1.

**Table 1.** Examples of basic part $S_{basic}$ and component-enhanced part $S_{cf}$.

| $S_{basic}$ | $S_{cf}$ |
| --- | --- |
| An Asian woman in a crowd is not carrying a black bag | woman not carrying bag |
| A man attacks a woman | man attacks woman |

Based on this idea, the overall architecture of the CF-BERT is shown in Figure 2. A similar network structure as that in SBERT [32] is used to design the CF-BERT model.

According to Figure 2, the two sentences A and B are passed to the CF-BERT model to generate fixed-sized sentence embeddings $emb\_SA_{basic}$ and $emb\_SB_{basic}$ as the basic components of A and B, respectively (described in Section 3.2). Subsequently, the CF-BERT performs the dependency parsing of sentences A and B to obtain the component-enhanced part with their relations (described in Section 3.3).

By the same way as the basic components, sentence embeddings emb_$SA_{cf}$ and emb_$SB_{cf}$ for the component-enhanced parts of A and B are also generated. Next, the weight factor $W_{cf}$ is introduced to adjust the ratio of the component-enhanced part embeddings $emb\_S_{cf}$ to the basic part embeddings $emb\_S_{basic}$ to generate the complete sentence representation. A complete sentence representation $emb\_S$ (fixed-sized vector) is finally expressed as follows:

$$emb\_S = emb\_S_{cf} * W_{cf} + emb\_S_{basic} \tag{1}$$

when $W_{cf} = 0$, CF-BERT equals SBERT.

The best weight factor $W_{cf}$ is obtained through a grid search method in the experiment (described in Section 5). Finally, we use the complete sentence representations from the previous step as the input of the last output layer. The output layer of the model can be changed according to the specific NLP tasks. This study investigates semantic textual similarity. We focus on textual similarity tasks only in this section and then evaluate the broader utility of our sentence representations on entailment classification in the next section.

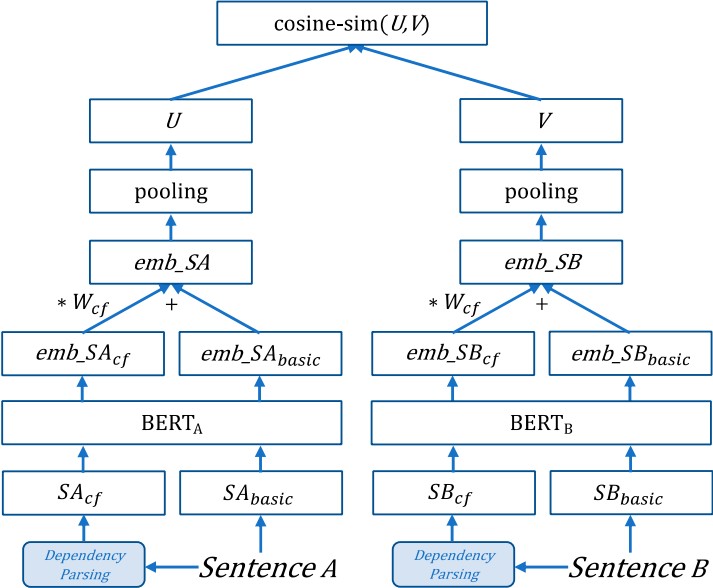

**Figure 2.** The overall architecture of the component focusing bidirectional encoder representations from transformers (CF-BERT) model (with semantic textual similarity tasks). The two BERT networks have tied weights (Siamese network structure).

### 3.2. Details of the CF-BERT Model with Similarity Tasks

BERT sets new state-of-the-art performance on semantic textual similarity. Before a sentence pair is entered into the BERT [12] network, a [CLS] token is added to the header, and a [SEP] token is added to separate the two sentences. Then taking the output of BERT, the [CLS] token embedding $C$ performs the similarity task (*sigmoid* ($CW^T$)), where $W$ represents the layer weight. However, $C$ represents the embedding of this sentence pair, and the embedding of each sentence cannot be calculated separately. SBERT [31] fine-tunes BERT in the manner of a Siamese network structure. The last layer makes cosine/Manhattan/Euclidean discrimination, which enables BERT to learn a proper sentence embedding suitable for similarity tasks. We introduce the CF-BERT model based on SBERT via component focusing in detail.

### 3.2.1. Training Process of CF-BERT

As shown in Figure 2, there are two networks, BERT$_A$ and BERT$_B$, which each process one of the sentences in a given pair; we solely focus on Siamese architectures with tied weights such that

$BERT_A = BERT_B$ in this work. First, we use the BERT model to map tokens in a sentence to the output embeddings from BERT (word_embedding_model). The next layer in our model is a pooling_model. We experiment with three pooling strategies: using the output of the [CLS] token, computing the mean of all output vectors (MEAN-strategy), and computing a max-over-time of the output vectors (MAX-strategy). When trained with different tasks, we observe that the pooling strategy has a large impact. There, the MEAN strategy performs better than MAX or [CLS] token strategy. Therefore, the default configuration is MEAN.

We performed dependency parsing to obtain the component-enhanced parts of A and B, namely, $SA_{cf}$ and $SB_{cf}$. The basic part $SA_{basic}$ and the component-enhanced part $SA_{cf}$ of sentence A are now passed first through the word embedding model, using the $BERT_A$ model to map tokens in a sentence to the output embeddings from $BERT_A$, namely, $emb\_SA_{basic}$ and $emb\_SA_{cf}$. The final sentence representation $emb\_SA$ of sentence A is calculated using Equation (1). $emb\_SA$ is then passed through the pooling_model to obtain fixed-sized sentence vectors $U$. Using the same process, sentence B is finally expressed as sentence vectors $V$. We used a batch-size of 16 and 4 epochs, an Adam optimizer with a learning rate $2e^{-5}$, and a linear learning rate warm-up over 10% of the training data.

Given a sentence pair and a gold similarity score (either between –1 and 1 or between 0 and 1), computes the cosine similarity between the sentence embeddings $U$ and $V$ and minimizes the mean squared error (mse) loss. The cosine-similarity is embedded in the loss layer, and then back-propagated. The previous step returns the propagation loss to calculate the gradient. The weights of the Siamese network are then updated by using the optimizer. We ran our experiments also with negative Manhattan and negative Euclidean distances as similarity measures, but the results for all approaches remained roughly the same.

When training is complete, the cosine similarity between all sentence pairs is computed and correlation to the gold scores is computed. The experiment automatically saves a model to embed sentences for the task. By loading the model, we can obtain the semantically meaningful embeddings of each sentence corresponding to the task.

### 3.2.2. Appropriate Choice of $BERT_A$ and $BERT_B$

The official Google artificial intelligence (AI) team provides a variety of pre-trained BERT models, for different languages and different model sizes. SBERT experiments with two setups: only training on specific NLP tasks, or first training on NLI, then on specific NLP tasks. SBERT observes that the latter strategy leads to the improvement of several points. Therefore, $BERT_A$ and $BERT_B$ are instantiated from "bert-base-nli-mean-tokens", which represents the BERT-base model fine-tuned with mean-tokens pooling on NLI, and "bert-large-nli-mean-tokens", which represents BERT-large with mean-tokens pooling. " BERT-large" has more parameters, deeper network layers, and a larger hidden size than "BERT-base", and can achieve better performance.

### 3.3. Component-Enhanced Part

In addition to the basic component, the component-enhanced component contains supplement information in the sentence representation in this paper. Compared with words, sentences have more complex structures, and a variety of relations is observed among words in the same sentence. From the knowledge of linguistics [38], every sentence is composed of crucial components (subject, predicate, object, etc.) and modifiers (attribute, adverbial, complement, etc.). The crucial components play a major role in the sentence, and the modifiers play a secondary role. Therefore, the subject, predicate, and object of a sentence are extracted for the component-enhanced part in this paper.

To learn the component-enhanced part of the sentence, the Stanford Parser, which is a natural language parser that determines the grammatical structure of sentences, is used. The Stanford Parser can find dependency relations information between words in sentences and output it in Stanford dependency format, including a directed graph shown in Figure 1. There are approximately 50 dependency relations such as noun subject (nsubj), direct object (dobj), and indirect object (iobj).

To extract the subject, predict, and object, we label the words that have the dependency relations including "subj", which identifies the subject of a sentence, and dependency relations including "obj", which identifies the predicate and object of a sentence. Details are shown in Table 2.

After traversing all dependency relations of a sentence, if "subj" and "obj" do not exist (this is relatively rare), we cannot obtain the subject-predicate-object in the above manner. We assume that all nouns are subject and object, and verbs are predicates for this sentence. To prevent the $S_{cf}$ part from being empty, we consider traversing all the words with the part of speech "NN" as the subject and object, and the words with the part of speech "VB" as the predicate, details shown in Table 2. In text tasks, sentences containing words like "no" or "not" can have a huge impact on the results of the experiment. So as long as words (like "not", "no") appear in the sentence, we also consider it in $S_{cf}$. The parsed words are arranged according to the dependency relations of the subject, predicate, and object. Therefore, the subject-predicate-object relations are also contained in $S_{cf}$. The component-enhanced part is regarded as a sentence which to feed into the sentence model, and the embedding of this part is obtained.

**Table 2.** All the dependency relation including subj and obj, parts of speech including VB (verb) and NN (noun) defined by Standford Parser and what they stand for.

| subj | obj | VB | NN |
|---|---|---|---|
| **csubj:** clausal subject<br>**csubjpass:** clausal passive subject | **dobj:** direct object<br>**iobj:** indirect object | **VB:** verb, base form<br>**VBD:** verb, past tense | **NN:** noun, singular or mass<br>**NNs:** noun, plural |
| **nsubj:** nominal subject | **pobj:** object of a preposition | **VBG:** verb, gerund/present | **NNP:** noun, proper noun, singular |
| **nsubjpass:** passive nominal subject<br>**xsubj:** controlling subject | | **VBN:** verb, past participle<br>**VBP:** verb, non-3rd ps. sing. present<br>**VBZ:** verb, 3rd ps. sing. present | **NNPS:** proper noun, plural |

The results produced by the Stanford Parser are listed in Table 3 (Bold represents what needs to be addressed in the dependency parsing.). In one triple "('carrying', 'VBG'), 'nsubj', ('woman', 'NN')", terms 'VBG' and 'NN' mean the parts of speech of the corresponding words; while 'nsubj' denotes the dependency relationship between 'carrying' and 'woman', in which 'woman' is the noun subject of 'carrying', 'man' is the noun subject of 'attacks'.

**Table 3.** Results produced by the Stanford Parser.

| An Asian Woman in a Crowd Is Not Carrying a Black Bag | A Man Attacks a Woman |
|---|---|
| (('carrying', 'VBG'), **'nsubj'**, ('woman', 'NN')) | (('man', **'NN'**), 'det', ('A', 'DT')) |
| (('woman', 'NN'), 'det', ('An', 'DT')) | (('man', 'NN'), 'dep', ('attacks', **'NNS'**)) |
| (('woman', 'NN'), 'amod', ('Asian', 'JJ')) | (('man', 'NN'), 'dep', ('woman', **'NN'**)) |
| (('woman', 'NN'), 'nmod', ('crowd', 'NN')) | (('woman', 'NN'), 'det', ('a', 'DT')) |
| (('crowd', 'NN'), 'case', ('in', 'IN')) | (('man', 'NN'), 'det', ('A', 'DT')) |
| (('crowd', 'NN'), 'det', ('a', 'DT')) | |
| (('carrying', 'VBG'), 'aux', ('is', 'VBZ')) | |
| (('carrying', 'VBG'), 'neg', (**'not'**, 'RB')) | |
| (('carrying', 'VBG'), **'dobj'**, ('bag', 'NN')) | |
| (('bag', 'NN'), 'det', ('a', 'DT')) | |
| (('bag', 'NN'), 'amod', ('black', 'JJ')) | |

## 4. Experiment

### 4.1. Datasets and Tasks

Two widely used datasets are employed in the experiment: Sentences Involving Compositional Knowledge (SICK) dataset and the Semantic Textual Similarity Benchmark (STS-B) dataset.

The SICK dataset was published in SemEval-2014 [39] Task 1 and is composed of two tasks: the Entailment Classification task SICK-E (entailment) and Semantic Textual Similarity task SICK-R

(relatedness). The dataset contains 9927 (4500 for train set/4927 for test set/500 for development set) pairs of sentences. Each sentence pair is annotated with a relatedness label $\in [1,5]$ corresponding to the average relatedness judged by ten different individuals; each of the SICK sentence pairs has also been labeled as one of three classes: entailment, contradiction, or neutral, which are to be predicted for the test examples.

The STS brings together the English data from the SemEval Semantic Textual Similarity tasks between 2012 and 2017. It was published to provide a standard benchmark to evaluate various semantic representation models. The dataset includes 8628 sentences pairs that are divided into train (5749), development (1500), and test (1379). STS aims to measure the degree of equivalence in meaning or semantics between a pair of sentences. The evaluation consists of human annotated English sentence pairs, scored on a scale of 0 to 5 to quantify the similarity of meaning, with 0 being the least, and 5 the most similar.

We provide a description and sample instances of these datasets and tasks in Table 4.

**Table 4.** Downstream tasks description and samples.

| Dataset | Task | Sentence A | Sentence B | Output |
|---------|------|-----------|-----------|--------|
| Sentences Involving Compositional Knowledge Semantic Relatedness (SICK-R) | To measure the degree of semantic relatedness between sentences from 0 (not related) to 5 (related) | A woman with a ponytail is climbing a wall of rock. | The climbing equipment to rescue a man is hanging from a white, vertical rock. | 1.8 |
| Sentences Involving Compositional Knowledge Entailment (SICK-E) | To measure semantic in terms of entailment, contradiction, or neutral | The dog is snapping at some droplets of water. | The dog is not snapping at some droplets of water. | Contradiction |
| Semantic Textual Similarity Benchmark (STS-B) | To measure the degree of semantic similarity between two sentences from 0 (not similar) to 5 (very similar) | A woman picks up and holds a baby kangaroo. | A woman picks up and holds a baby kangaroo in her arms. | 4.6 |

### 4.2. Evaluation Metrics

In the Semantic Textual Similarity task, there are two evaluation metrics: Pearson correlation and Spearman correlation. Although Pearson correlation is the official ranking basis for the semantic textual similarity tasks, we use Spearman correlation as an auxiliary verification at the same time. The goal of the task is to obtain the largest possible value of both for the test set.

In the Entailment Classification, the models are evaluated in terms of classification accuracy. The goal of the task is to obtain a higher accuracy for the test set.

### 4.3. Evaluation—Semantic Textual Similarity (STS)

State-of-the-art methods often learn a complex similarity function that maps sentence embeddings to a similarity score. However, these similarity functions work pair-wise and due to the combinatorial explosion, are often not scalable if the collection of sentences reaches a certain size. Instead, we use cosine similarity to compare the similarity between two sentence embeddings.

We evaluated the performance of Universal Sentence Encoder (USE), USE with component focusing, SBERT (first training on NLI, then training on the specific dataset), and CF-BERT (first training on NLI, then training on specific dataset via component focusing) on common STS tasks. We implemented two kinds of CF-BERT based on two pre-trained BERT models, namely, CF-BERT$_{\text{BASE}}$ and CF-BERT$_{\text{LARGE}}$. The weight factor $W_{cf}$ sets the value to 0.2 in the two tasks (described in the next section).

We ran our experiments both with Pearson correlation and Spearman correlation as similarity correlation measures. The results are depicted in Table 5. Pearson correlation ($r$) and Spearman

correlation ($\rho$) between the cosine similarity of sentence representations are for two Semantic Textual Similarity (STS) tasks. Performance is reported by convention as $r \times 100$ and $\rho \times 100$. The first group of results are the top four SemEval-2014 submissions [40] for SICK-R, the second are more recently proposed methods (including average methods), and the third contains BERT-based methods. The results in black bold represent the best of all methods.

**Table 5.** Test set Pearson correlations ($r$) and Spearman correlations ($\rho$) for STS task.

| | Models | SICK-R | | STS-B | |
|---|---|---|---|---|---|
| | | $r$ | $\rho$ | $r$ | $\rho$ |
| Top Four SemEval-2014 submissions for SICK-R [40] | ECNU_run1 | 82.8 | 76.89 | | |
| | StanfordNLP_run5 | 82.72 | 75.59 | — | |
| | The_Meaning_Factory_run1 | 82.68 | 77.22 | | |
| | UNAL-NLP_run1 | 80.43 | 74.58 | | |
| Recently proposed methods (2015–2018) | Avg. GloVe embeddings [2] | 69.43 | 53.76 | 61.23 | 58.02 |
| | Avg. BERT embeddings [28,29] | 71.4 | 58.4 | 51.17 | 46.35 |
| | Universal Sentence Encoder (USE) [27] | 82.65 | 76.69 | 75.53 | 74.92 |
| | USE (with CF $W_{cf} = 0.2$) (our) | 83.51 | 77.38 | 76.22 | 75.54 |
| | Skip-thought [23] | 85.84 | 79.16 | 77.1 | 76.7 |
| | Infersent [24] | 88.4 | 83.1 | 75.8 | 75.5 |
| BERT-based methods | BERT$_{BASE}$ [12] | 87.9 | 83.3 | 87.1 | 85.8 |
| | SBERT$_{BASE}$ [31] | 87.81 | 83.26 | 87.04 | 85.76 |
| | CF-BERT$_{BASE}$ (with CF $W_{cf} = 0.2$) (our) | 88.39 | 83.77 | 87.58 | 86.29 |
| | BERT$_{LARGE}$ [12] | 88.9 | 84.2 | 87.6 | 86.5 |
| | SBERT$_{LARGE}$ [31] | 88.78 | 83.93 | 87.41 | 86.35 |
| | CF-BERT$_{LARGE}$ (with CF $W_{cf} = 0.2$) (our) | 89.29 | 84.35 | 87.83 | 86.82 |

Table 5 shows that directly using the output of BERT embeddings leads to rather poor performance even worse than computing average word embeddings (GloVe) sometimes. Using the SBERT (described Siamese network structure and fine-tuning mechanism substantially) improves the correlation, outperforming both Skip-thought and Universal Sentence Encoder. SBERT is slightly weaker than BERT in general. Maybe because by comparing CF-BERT to BERT, there is less interaction of the sentence pairs.

The results from Table 5 indicate that all methods with the addition of component focus have improved performance compared to methods without component focusing. USE with component focusing leads to an improvement of 0.7–0.9 points of two tasks. For SICK-R test set, the Pearson correlations of CF-BERT$_{BASE}$ and CF-BERT$_{LARGE}$ are improved by approximately 0.58 and 0.51 points compared to SBERT$_{BASE}$ and SBERT$_{LARGE}$. The Spearman correlations of CF-BERT$_{BASE}$ and CF-BERT$_{LARGE}$ are improved by approximately 0.51 and 0.42 points compared to SBERT$_{BASE}$ and SBERT$_{LARGE}$. The results for STS-B test set remained roughly the same. The Pearson correlations lead to an improvement of 0.54 and 0.42 points to CF-BERT$_{BASE}$ and CF-BERT$_{LARGE}$. The Spearman correlations lead to an improvement of 0.53 and 0.47 points to CF-BERT$_{BASE}$ and CF-BERT$_{LARGE}$. Our methods easily exceed the first four results. Moreover, the performance of CF-BERT is slightly better than that of BERT; CF-BERT$_{LARGE}$ is able to outperform the other sentence models.

## 4.4. Evaluation—Entailment Classification

To evaluate the broader utility of our sentence representations, we leverage them for a different application: the SemEval-2014 textual entailment task (SICK-E). For this task, we solely rely on the same representations learned for predicting semantic textual similarity (SICK-R), and simply apply standard learning methods to do the entailment classification. From the sentence representations emb_SA, emb_SB of each pair of sentences, we first pass the pooling_model to get sentence embeddings U, V of sentence A and B, we then compute the following simple feature: element-wise (absolute) differences |U − V|. Using only this feature label, we train a logistic regression to classify the entailment labels. The logistic regression classifier is trained on various tasks in a 10-fold cross-validation setup

and the prediction accuracy is computed for the test-fold. The results are depicted in Table 6. Accuracy (*100) is reported for the results. The first group of results are the top four SemEval-2014 submissions for SICK-E, the second are more recently proposed methods (including average methods), and the third is BERT-based methods. The results in black bold represent the best of all methods.

**Table 6.** Test set accuracy for the SICK semantic entailment classification.

| | Models | SICK-E |
|---|---|---|
| Top Four SemEval-2014 submissions for SICK-E [40] | Illinois-LH_run1 | 84.6 |
| | ECNU_run1 | 83.6 |
| | UNAL-NLP_run1 | 83.1 |
| | SemantiKLUE_run1 | 82.3 |
| Recently proposed methods (2015–2018) | Avg. GloVe embeddings [2] | 74.3 |
| | Avg. BERT embeddings [28,29] | 78.5 |
| | Universal Sentence Encoder (USE) [26] | 83.5 |
| | USE (with CF $W_{cf}$ = 0.2) (our) | 84.3 |
| | Skip-thought [23] | 82.3 |
| | Infersent [24] | 86.3 |
| BERT-based methods | BERT$_{BASE}$ [12] | 85.8 |
| | SBERT$_{BASE}$ [31] | 85.8 |
| | CF-BERT$_{BASE}$ (with CF $W_{cf}$ = 0.2) (our) | 86.3 |
| | BERT$_{LARGE}$ [12] | 87.2 |
| | SBERT$_{LARGE}$ [32] | 87.1 |
| | CF-BERT$_{LARGE}$ (with CF $W_{cf}$ = 0.2) (our) | 87.5 |

Table 6 shows that all methods with the addition of component focus have improved performance compared to methods without component focusing. USE improves performance with around 0.8 points, CF-BERT$_{BASE}$ improves performance with about 0.5 points, and CF-BERT$_{LARGE}$ improves performance with about 0.4 points. The performance of CF-BERT is slightly better than that of BERT; CF-BERT$_{LARGE}$ outperforms all other textual-entailment systems.

*4.5. Overall Evaluation*

It appears that the sentence embeddings from SBERT capture sentence information well: we observe significant improvements for all tasks (STS and entailment classification) in comparison to Skip-thought and Universal Sentence Encoder (USE). Dependency parser is a pre-processing process in CF-BERT and only runs once. In addition, the training process of CF-BERT is roughly equivalent to SBERT. The downstream NLP task datasets we take are small datasets, so we do not discuss time and memory in detail here. The experiential results demonstrate the universality of component focusing, all models with component focusing achieve a higher score in NLP tasks than those without. Therefore, the addition of component focus plays a role in optimizing sentence representation, and our model can learn more meaningful sentence representations.

However, with the same component focus method, the performance growth from USE and CF-BERT$_{BASE}$ to CF-BERT$_{LARGE}$ is diminishing. We speculate that this may be because the performance of the USE model is intrinsically inferior to the CF-BERT model. In other words, the CF-BERT models are powerful enough because CF-BERT has fine-tuned for specific tasks and can capture more semantic information than USE. Therefore, the increase in results obtained by enhancing the component focus part may also be limited. Similarly, CF-BERT$_{LARGE}$ is closer to the performance ceiling than CF-BERT$_{BASE}$, so even if other useful information of representation is added, the performance improvement of CF-BERT$_{LARGE}$ may be smaller.

## 5. Discussion: The Impact of Weight Factor $W_{cf}$ on Model Performance

As mentioned before, weight factor $W_{cf}$ is used to adjust the weight of the component-enhanced part so that the best sentence representation is obtained by Equation (1). This section selects a grid

search to study the impact of different weight factors on model performance and describes how to find the $W_{cf}$ corresponding to the best performance.

Grid search is used for hyperparametric optimization to improve model performance by optimizing the optimal combination of hyperparameters. We only need to determine one hyperparameter $W_{cf}$ in this paper. Firstly, a list of values is set for the hyperparameter of $W_{cf}$, and then the computer goes through each to evaluate the performance and select the best one.

We first chose $W_{cf} \in \{2.0,3.0,4.0,5.0,6.0,7.0,8.0,9.0,10.0\}$ to study the influence of the weight factor on model performance, then $W_{cf}$ is limited to the range of 0 (without component focusing) to 1.0, with the increasing rate of 0.1 ($W_{cf} \in \{0, 0.1,0.2,0.3,0.4,0.5,0.6,0.7,0.8,0.9,1.0\}$). We changed $W_{cf}$ and performed SICK-R, SICK-E, and STS-B on CF-BERT$_{BASE}$ and CF-BERT$_{LARGE}$. We did not observe a significant difference between CF-BERT$_{BASE}$ and CF-BERT$_{LARGE}$. Experimental results of CF-BERT$_{LARGE}$ on SICK-R and STS-B are displayed in Figure 3 and Table 7 (only Pearson correlation (*100) is displayed).

We find that the first experiment brings terrible results. Performance is much worse than the models without component focusing. When $W_{cf} > 6.0$, all the performance degradation exceeds six points (we did not show $W_{cf} \in \{7.0, 8.0,9.0,10.0\}$). This demonstrates that component focus can be used as a means of feature supplement rather than dominance. As shown in Figure 3, when the weight factor $W_{cf}$ is in the range of 0.1 to 0.5, the score first increases and then decreases, but all of the performance is improved compared to the case of $W_{cf} = 0$. Further increasing the value of $W_{cf}$ would bring negative feedback. When $W_{cf}$ exceeds 0.5, the performance decreases as $W_{cf}$ increases, and the performance is even lower than the model without component focusing ($W_{cf} = 0$). Experiments always get the best results when $W_{cf}$ reaches around 0.2. The results shown in Table 7 in black bold represent the best of all models and those underlined represent the second-best results.

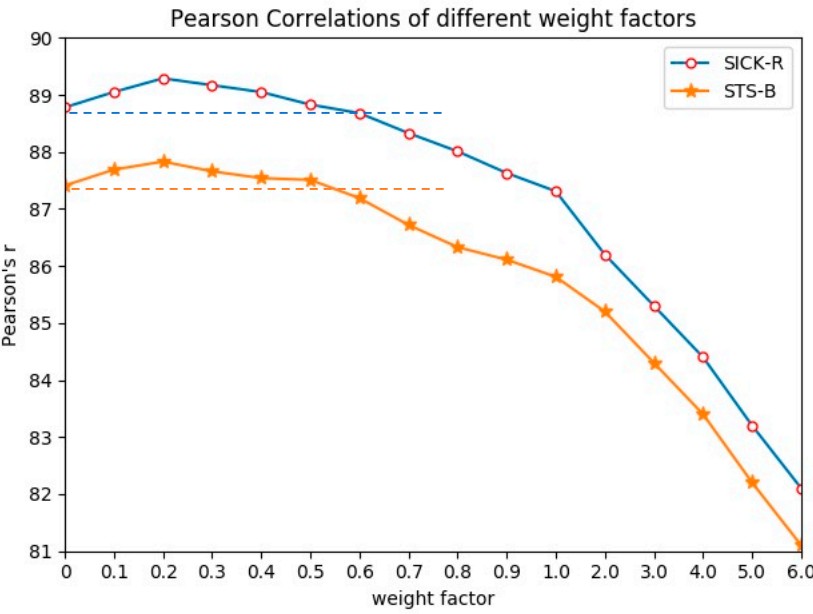

**Figure 3.** Pearson's (*r*) of different weight factors. The horizontal axis is the range of weight factor $W_{cf}$, and the vertical axis is the correlation coefficient. The two auxiliary horizontal lines represent the value when $W_{cf} = 0$ (method without CF) to facilitate comparison with all the results. The variation of similarity between STS and SICK-R is roughly consistent.

**Table 7.** Pearson correlations of different weight factors for CF-BERT$_{LARGE}$.

| $W_{cf}$ | 0 | 0.1 | 0.2 | 0.3 | 0.4 | 0.5 | 0.6 | 0.7 | 0.8 | 0.9 | 1.0 | 2.0 | 3.0 | 4.0 | 5.0 | 6.0 |
|---|---|---|---|---|---|---|---|---|---|---|---|---|---|---|---|---|
| SICK-R | 88.78 | 89.05 | **89.29** | <u>89.17</u> | 89.05 | 88.83 | 88.68 | 88.33 | 88.01 | 87.63 | 87.31 | 86.22 | 85.23 | 84.34 | 83.12 | 82.01 |
| STS-B | 87.41 | <u>87.69</u> | **87.83** | 87.66 | 87.54 | 87.51 | 87.19 | 86.72 | 86.33 | 86.11 | 85.81 | 85.18 | 84.26 | 83.38 | 82.17 | 81.10 |

## 6. Conclusions

In this paper, we introduce a component focus sentence representation method based on modifying BERT with a Siamese network (CF-BERT). Due to focusing on crucial components (mainly from subject, predicate, and object) with their dependency relations, there is a way to interpret the sentence embedding in depth in our model. Experimental results over two different tasks show that the model outperforms other sentence models. To implement this idea, CF-BERT divides a sentence representation into two parts: a basic part (the complete sentence) and a component-enhanced part, which contains the crucial components of a sentence with their relations acquired by dependency parsing. Both parts can be mapped into fixed-sized sentence vector embeddings by using CF-BERT. Additionally, the weight factor is introduced to adjust the ratio of the two parts to obtain the complete semantically meaningful sentence representation. The results indicate that the addition of a component-enhanced part within a certain range improves the sentence representation. This confirms that the method using component focus improves the representation of sentences, reduces the impact of noisy words on sentence meanings, and then performs effective NLP tasks.

In future work, we will specifically analyze different downstream tasks and leverage other lexical and syntactic information, such as the part-of-speech (POS). Words with different POS are of varying importance in a sentence (for example, adverbs are much more crucial in sentiment analysis tasks). We can assign the same weight to words with the same POS in a sentence. Each POS corresponds to a different weight that can be automatically trained by the model.

**Author Contributions:** Writing—original draft preparation, X.Y.; methodology, X.Y.; conceptualization, W.Z. and T.Y.; data curation, S.L.; writing—review and editing, W.Z. and W.Z.; funding acquisition, W.Z. and W.Z. All authors discussed the results and contributed to the final manuscript. All authors have read and agreed to the published version of the manuscript.

**Funding:** The work of this paper is supported by National Natural Science Foundation of China (No. 61572434) and Shanghai Science and Technology Committee (No. 19DZ2204800), Program of Shanghai Municipal Education Commission (No. 2019-01-07-00-09-E00018) and the National Key R&D Program of China (No. 2017YFB0701501).

**Conflicts of Interest:** The authors declare no conflicts of interest.

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
