# Peer review of "Improving Sentence Representations via Component Focusing"

_applsci, doi:10.3390/app10030958_

Round 1

Reviewer 1 Report

While I appreciate the effort of the authors who made notable improvements to the manuscript compared to its first version, the current form of the paper is still not ready for publication.

The main problem with the paper is that the proposed method is not described clearly.
First, the difference between S and Sbasic is not explained (the examples for S and Sbasic provided in Table 1 are the same sentences).
Then, the method is described using examples of two sentences (A and B) which seems adequate, yet only for the task of semantic textual similarity measurement. Either, the entire paper should be rewritten, focusing on textual similarity only, or the presentation of the method should be rewritten, by first explaining its workings, and then its application to specific NLP tasks (such as textual similarity measurement).

I also disagree with authors' response regarding the lack of information on time and memory consumption of the proposed method ("Everyone uses a different device. In the same code, some people use CPU and the other may use GPU, it's difficult to compare the metrics of time and memory consume").
It is the role of the authors to test their own method against those it is supposed to replace (on their own hardware). Otherwise, we cannot know if the attained improvement in effectiveness is worth the cost of extra processing time.

The purpose of some statements is unknown, e.g.:
r. 176-7: This study investigates 2 different tasks: semantic textual similarity (mainly for this paper) and entailment classification" (what does "mainly for this paper" is supposed to mean?)
r. 252-4: "We randomly extract 100 sentences and compare the subject-predicate-objects obtained by the manual annotation and the dependency parsing and find that the above operations can basically get the correct results" (are the authors reporting their own verification of Stanford Parser results' quality?)

The English of the manuscript has been improved, yet it still requires further correction of grammar and style, e.g.:

r. 98: "Learning representations of sentences, also called sentence embeddings, are a well-studied" (are->is)
r. 105: "In methods of neural networks" (wording)
r. 139: "Tai et.al (2015) [33]design" (lack of whitespaces)
r. 203/4: "emb_SA is then through the pooling_model to obtain" (lack of verb)
r. 210: "Returns the propagation loss to calculate gradient." (no subject)
r. 210: "Using the optimizer to update the weights Siamese network." (no verb and no subject)
r. 218: "The official provides a variety of pre-trained BERT models" (who is the official?)

The manuscript needs also further editing, e.g.:
r. 23: acronym BERT is introduced without definition.
r. 48: "is essentially an attention structure that can directly obtain global information"
- here, similarly, the concept of attention is referenced without proper introduction.

Reviewer 2 Report

This paper proposes a new representation for sentences: CF-BERT. This is a relevante topic and the idea behind it is quite interesting: to enrich BERT with more linguistic based information; in this paper the authors propose to use subj-verb-obj and/or noun expressions as "component focus".   However it is not clear if the goal of this paper is to propose a new sentence representation technique (CF-BERT), which can be used in many NLP taks, or if the focus is the semantic similarity and textual entailment task. This should be quite clear throug all the paper and, in my opinion, all the paper should be revised accordingly. For instance, if the goal is to propose a new sentence represenatation technique, then Figure 2 is not adequate because it represents an architecture for semantic similarity tasks. If, on the other hand, the goal of the paper is the semantic similarity task, then the title, abstract, introduction should be revised.   Moreover, a more formal definition of how Sdep is obtained should be presented. The input is the dependency parser output but it is not clear which words are selected (seems to be the *subj, *obj, vb* and, in some situations, the nominal expressions and the "neg" terms). Why not use also the *mod? (a "black cat" is different from a "white cat"). Did you evaluate this and other possibilities?   An English revision to the document should also be made.  

Reviewer 3 Report

This paper proposes a method to incorporate part-of-speech information to improve sentence representation in BERT-based natural language processing.  The proposed method extracts the subject-predicate-object relationship from sentences and enhances their weights in the BERT training process. The proposed method is compared with other similar approaches to text similarity detection tasks and showed comparable or slightly better performance. The authors also discussed the selection of the weight coefficient.

Overall, I find the paper clear and easy to read. Except for several places that need further clarification, I found the paper sound and interesting.

The introduction clearly states the motivation of the paper. I like that the authors connect to two prior papers which suggested the importance of the word relationship. To make the theoretical background more solid, I think a discussion or summarization of more papers on word dependency would be even more helpful.

Section 3 gives the details of the CF-BERT. Overall, I understand how the method works. However, I still have questions about why the model is designed in such a way and whether alternatives can be used. For example, the final embedding depends on the original sentence and the enhanced sentence. Why the original sentence is still used? What are the roles of Sbasic and Sdep in calculating the result? The crucial sentence information is limited to the subject, predicate, and object. How crucial information categories are determined? Why words with other POS tags are not used? If other categories of words are used, how it affects the result?

Some of the configuration choices are not clearly informed and discussed. The authors experimented with three pooling strategies but picking MEAN as default without informing the reason. Line 238-248 discusses word relationships, but I don’t understand what it means by the dependency does not exist. Line 252-254 does not state where the 100 sentences are extracted from. In this section, I also don’t understand what “basically get correct results” mean. To what degree the annotations are not correct?

The experiment compares the proposed method with prior methods on benchmark datasets. CF-BERT has comparable performance with BERT and SBERT. In this section, I think more discussion on the practice will be helpful. What are the practical advantages of the proposed method? When people should use CF-BERT rather than BERT and SBERT when conducting similar NLP tasks?

The last problem with the current version is the short future work section. I think a more insightful discussion on how other lexical and syntactic information can be used to improve the results is needed.

Reviewer 4 Report

In this paper, the authors propose a modified BERT sentence representation method with a component-focusing feature. The authors aim in improving the sentence representation by reducing the noise and focusing on “crucial components of a sentence”: subject, predicate and object components.
The main formula of the proposed approach is given in Equation (1) with a Section 5 describing various valued for the weight factor.

Some observations:
1. The authors should explain why they use “*” in the dependency relations names and POS data.
2. The authors should not use notations BERT_A and BERT_B, these are confusing, giving the idea we have two models, instead we have the same BERT model applied on two sentences A and B
3. The main (and only) equation of the model applied by the authors, that is, Equation 1, must be re-written according to the component-enhanced approach defined by the authors
4. The paper is hard to read due to many language errors, please revise! such as:
- a “which” is missing in the abstract,
- “which each process” in section 3.2.1;
- in the phrase “Unlike RNN, which requires gradual recursion to obtain global information, and unlike CNN, which can only obtain local information. Transformer outperforms RNN and CNN in machine translation tasks.” remove “.”
- “is then through the pooling_model to obtain fixed sized sentence vectors” is what?
- “we label the words have the dependency relation” the words THAT have

Round 2

Reviewer 1 Report

Again, I appreciate the authors' effort, however not all weaknesses pointed to in the previous round of review were adequately addressed, and the paper is still too chaotic to be considered ready for publication.
Regarding the authors' explanations:

"(...)Therefore, we have taken a simple way to cover the global information is to let Sbasic = S."

In this context, the authors' writing is completely misleading:
"the sentence representation consists of two parts, and to make it easy to describe a sentence representation that consists of two parts, we give the raw sentence text of the two parts names ?????? and ???", as talking about two parts implies that one needs both of them to obtain the full sentence, whereas in fact, ?????? is the full sentence.

Therefore, I still do not see any reason for defining Sbasic as a separate notion to S.

Moreover, the presented formula S_basic= S-S_dep is mathematically wrong, as the difference S-S_dep does not equal S_basic (which, as explained by the authors is equal to S).

"(...)Our paper focuses on the similarity task", "We have updated the manuscript by modifying some confusing expressions in section 3 Method."

The Introduction still does not provide any insight of what follows as all it says about the paper structure and content is:
"design of the method is described in section 3. Then, we explain the data sources, experimental details, experimental results, evaluation and discussion of the experiment in section 4, 5."

This does not conform accurately to the actual contents of these sections.

According to the abstract ("we propose a sentence representation model"), the key contribution is the representation (note, in Introduction, the authors write: "The key contribution of this paper is focusing on the crucial components of a sentence" - I suggest this to be rewritten as well, as "focusing" is a vague contribution).
Section 3 still describes both the representation and its application for sentence comparison.
Once again, I suggest that either these two things should be decoupled (first, present the representation, then, describe how to use it), or the whole paper should be refocused on just sentence comparison (with an adequate change of the title, abstract and introduction [note the introduction has to be improved anyway]).

"There are two reasons why we do not explicitly mention time and memory in the paper"

While I agree with the authors that memory and time measurements do not necessarily have to be included in the paper (though it would increase its value), I still believe this aspect should at least be mentioned in the paper, so that it is obvious that the newly proposed method does not differ considerably in this regard to the existing solutions.

"We updated the manuscript by modifying the unknown statements."

Ok.

"We updated the manuscript by modifying the error places."

Ok.

Two new editing remarks:

6. Conclusions
=> should be:
6. Conclusion

and not all the references conform to the journal requirements, e.g. sometimes first names are given, sometimes only initials; sometimes proceedings titles are abbreviated, sometimes they are not etc.

Reviewer 2 Report

I believe the authors have answered in an adequate way to my comments.

Author Response

Thanks a lot for yours comment.

Reviewer 4 Report

I accept the paper in the present form, the study is improved from the presentation point of view.

Author Response

Thanks a lot for yours comment.

Round 3

Reviewer 1 Report

Thank you for the corrections. 

As all my points were addressed, I have no other remarks.

This manuscript is a resubmission of an earlier submission. The following is a list of the peer review reports and author responses from that submission.

Round 1

Reviewer 1 Report

The English of the manuscript is very poor. I strongly recommend the help of a native speaker in rewriting the paper. There are many style and grammar errors, even in the abstract: e.g. r. 17/18 "the most meaning of a sentence", r. 24 "To utilizing the performance"; there are also typos, e.g. Table 2: "clausual". The wording is so wrong that the text becomes incomprehensible, for instance: "Weight factor can decide basic part occupies the dominant position and component-enhanced part plays a supplement role" (r. 80-82). I have also no clue why section "Patents" is titled so.

The flow of reasoning is hard to follow. Section 3.1 gets into deep technical remarks without an adequate prior introduction. Table 5 presents some numbers, but it is not clear what they are.

In general, it is not clear from the paper what the authors actually do, and what are the benefits.
The claim (r. 26/27) "The experimental results show that performance is improved by expanding extra components and can outperform state-of-the-art results in different NLP tasks" makes no sense, as sentence representation is a state not a process, therefore cannot be discussed in terms of performance and can in no way outperform "different NLP tasks". I can only guess that the proposed sentence representation can help in performing specific tasks, but it should have been written that way, and various dimensions should be considered in the comparison (not only effectiveness but also the resources such as time and memory required to achieve the results).

I am far from saying that the authors contribution is without value or uninteresting, but the paper simply has to be rewritten entirely as in its current form it is unacceptable.